# Combining Solid and Liquid Biopsy for Therapy Monitoring in Esophageal Cancer

**DOI:** 10.3390/ijms241310673

**Published:** 2023-06-26

**Authors:** Florian Richter, Clara Henssen, Tim Alexander Steiert, Tobias Meissner, Anne-Sophie Mehdorn, Christoph Röcken, Andre Franke, Jan-Hendrik Egberts, Thomas Becker, Susanne Sebens, Michael Forster

**Affiliations:** 1Department of General and Thoracic Surgery, University Hospital Schleswig-Holstein Campus Kiel, 24105 Kiel, Germany; florian.richter@uksh.de (F.R.); anne-sophie.mehdorn@uksh.de (A.-S.M.); thomas.becker@uksh.de (T.B.); 2Institute of Clinical Molecular Biology, Kiel University, 24105 Kiel, Germany; clara.henssen2@uksh.de (C.H.); t.steiert@ikmb.uni-kiel.de (T.A.S.); a.franke@mucosa.de (A.F.); 3Department of Molecular and Experimental Medicine, Avera Cancer Institute, Sioux Falls, SD 57105, USA; tobias.meissner@avera.org; 4Department of Pathology, University Hospital Schleswig-Holstein Campus Kiel, 24105 Kiel, Germany; christoph.roecken@uksh.de; 5Department of Surgery, Israelitisches Krankenhaus Hamburg, 22297 Hamburg, Germany; j.egberts@ik-h.de; 6Institute for Experimental Cancer Research, Kiel University, University Hospital Schleswig-Holstein Campus Kiel, 24105 Kiel, Germany; susanne.sebens@email.uni-kiel.de

**Keywords:** EC, cell-free DNA, cfDNA, circulating tumor DNA, ctDNA, NGS, *SETBP1*, sequencing

## Abstract

Esophageal cancer (EC) has one of the highest mortality rates among cancers, making it imperative that therapies are optimized and dynamically adapted to individuals. In this regard, liquid biopsy is an increasingly important method for residual disease monitoring. However, conflicting detection rates (14% versus 60%) and varying cell-free circulating tumor DNA (ctDNA) levels (0.07% versus 0.5%) have been observed in previous studies. Here, we aim to resolve this discrepancy. For 19 EC patients, a complete set of cell-free DNA (cfDNA), formalin-fixed paraffin-embedded tumor tissue (TT) DNA and leukocyte DNA was sequenced (139 libraries). cfDNA was examined in biological duplicates and/or longitudinally, and TT DNA was examined in technical duplicates. In baseline cfDNA, mutations were detected in 12 out of 19 patients (63%); the median ctDNA level was 0.4%. Longitudinal ctDNA changes were consistent with clinical presentation. Considerable mutational diversity was observed in TT, with fewer mutations in cfDNA. The most recurrently mutated genes in TT were *TP53*, *SMAD4*, *TSHZ3*, and *SETBP1*, with *SETBP1* being reported for the first time. ctDNA in blood can be used for therapy monitoring of EC patients. However, a combination of solid and liquid samples should be used to help guide individualized EC therapy.

## 1. Introduction

Esophageal cancer (EC) has a dismal prognosis and is the sixth leading cause of cancer-related death [1]. For EC, multimodal neoadjuvant concurrent chemoradiotherapy (nCCRT) is increasingly used [2]. A convincing survival benefit has been demonstrated using nCCRT followed by surgery in patients with locally advanced EC [3]. Postoperative adjuvant chemotherapy has been shown to improve disease-free survival rates and to reduce the length of hospital stays in patients suffering from EC [4]. However, two different entities need to be considered: in the case of locally advanced and/or lymph-node-positive esophageal squamous cell carcinoma (ESCC), nCCRT is considered standard of care, followed by surgery. No adjuvant chemotherapy is applied. In the case of locally advanced and/or lymph-node-positive esophageal adenocarcinoma (EAC), perioperative chemotherapy, consisting of pre- and postoperative CCRT, is commonly used.

Despite curative intent, many patients develop recurrences and distant metastasis because current therapeutic concepts often fail. It has been reported that 40% to 50% of patients with supposedly localized EC suffer from recurrence or metastasis within two years after surgery [5,6], and the median survival after recurrence is only eight months [7]. Even early-stage tumors can relapse within two years after surgery.

Especially for advanced tumor stages, an additional diagnostic method is needed for the neoadjuvant chemotherapy evaluation to detect whether there is a treatment response or not. During neoadjuvant treatment, the disease often progresses, and curative treatment is no longer possible [8,9]. Thus, a large proportion of patients received ineffective chemotherapy over a long period of time, possibly with simultaneous radiation, both associated with considerable side effects. In these patients, early detection of treatment failure would allow an earlier change in therapy, associated with higher chances of curation. As another benefit, an early switch to an alternative therapy would avoid the side effects of an ineffective neoadjuvant treatment. Therefore, in addition to a tumor prognosis at an early stage, liquid-biopsy-based biomarkers could also represent a real-time assessment of the therapeutic response.

During tumor development and progression, EC cells acquire a malignant phenotype, leading to their dissemination from the primary tumor into the bloodstream and thereby to secondary sites [10]. These cells can be detected as biomarkers in peripheral blood, e.g., as circulating tumor cells (CTCs) [11,12]. In addition, cell-free DNA (cfDNA) which mainly stems from apoptotic cells [13] can be detected in blood [14]. The cfDNA consists of two fractions: one from healthy cells and one from tumor cells. The fraction from tumor cells can be identified by the presence of somatic mutations and is known as ctDNA.

In recent years, cfDNA detection with next-generation sequencing (NGS) has been established for different tumor entities [15,16,17], often using targeted deep sequencing of recurrently mutated genes. For several tumor entities, mutation detection in cfDNA is approved for tumor genetic examination. For example, RAS-mutation detection in colorectal cancer patients or EGFR-T790M companion diagnostics for osimertinib in lung cancer patients are reimbursable examinations in Germany. However, for EC, mutation detection in cfDNA has not been sufficiently studied for clinical implementation. So far, contemporary whole-genome and exome analyses of EC tumor tissue (TT) revealed a complex mutational landscape and identified significantly mutated genes, including *TP53*, *ZNF750*, *NOTCH1*, *FAT1*, and *NFE2L2* [18,19,20,21].

The objective of this study was to clarify the discrepancies in previous studies in order to evaluate the utility of cfDNA in liquid biopsies of EC patients for longitudinal monitoring of therapeutic responses and recurrence after resection. Only a few studies exist, some of which report that it is highly challenging to detect TT mutations in the plasma cfDNA of EC patients, compared to other cancer entities. Notably, Azad and colleagues examined plasma cfDNA from 45 EC patients and reported the median proportion of mutated circulating tumor DNA (ctDNA) in cfDNA before treatment to be 0.07% [22], which is significantly lower than in lung cancer (*p* = 0.009). They detected ctDNA in 60% of the patients before treatment, reporting ctDNA levels for ESCC to be seven-fold higher than for EAC and ctDNA levels for UICC IV (Union for International Cancer Control stage IV [23]) to be significantly higher than for UICC III. Liu et al. detected TT mutations in 38 of 60 ESCC patients (63%) and tissue-matching ctDNA in 30 of these 38 patients (78.9%), with a median variant allele frequency (VAF) of 0.76% [24]. This was 10-fold higher than in Azad et al.’s mixed EC cohort [22]. Ococks et al. detected pre-operative ctDNA in 37 of 75 resectable EAC patients (49%) with a median VAF of 0.52% [25], which is seven-fold higher than the 0.07% VAF reported by Azad and colleagues. Ococks et al. did not detect any VAF differences before and after neoadjuvant chemotherapy. Luo et al. performed exome sequencing of cfDNA from 11 ESCC patients’ pre-surgery and post-surgery plasma. They reported VAFs up to 100% in pre-surgery plasma and VAFs up to 60% in post-surgery plasma. They reported that ctDNA was detected in all patient samples (100%) but that sensitivity and specificity for individual mutations might be improved by deeper sequencing and the use of targeted enrichment [26]. Others reported detection rates for ctDNA ranging from 14% to 48% of EC patients when using NGS to detect EC TT mutations in plasma [27,28,29].

It is unclear whether the differences in EC ctDNA detection rate and EC cfDNA VAFs reported in the above studies can be explained by these or other confounders, such as patient selection bias. Therefore, we were motivated to investigate the detection rates of ctDNA in non-selected EC patients that represent the real-life diversity in our medical center. Further, we investigated the VAF levels before and after neoadjuvant treatment and at different follow-ups. Twenty-eight EC patients were prospectively recruited between 2019 and 2021 at the University Medical Center Schleswig Holstein. Our cohort consisted mainly of EAC cases, which according to Azad et al. are technically more challenging than ESCC with respect to ctDNA detection [22]. Blood samples in biological duplicates (two blood tubes) and TT samples from the primary site and metastatic sites were collected and sequenced using a broad panel of 127 cancer genes. Tumor tissue is accepted as the gold standard for detecting mutations. As the main conclusion, ctDNA can be detected before neoadjuvant therapy in a large proportion of EC patients (63%) with VAFs greater than 0.1% (median VAF 0.4%), which allows the monitoring of therapy response and clinical course in these patients.

## 2. Results

### 2.1. Characterization of the Analyzed Patient Cohort

Twenty-eight patients were recruited for this retrospective observational study, but seven patients had to be excluded (Figure 1). The studied patient collective consisted of 21 patients with a median age of 68 years (range 49–86 years). Out of the 21 patients, 17 (81%) were male and four (19%) were female. Nineteen (90.5%) patients presented with adenocarcinoma; two (9.5%) patients presented with squamous cell carcinoma. Thirteen (61.9%) patients received neo-adjuvant chemotherapy, whereas eight (38.1%) patients were untreated before surgery. Adjuvant treatment was administered to 10 (47.6%) patients in our cohort. Histopathological examination of the resection specimens classified one patient as stage I (4.8%) according to UICC, four as stage II (19%), ten as stage III (47.6%), and six patients were in stage IV (28.6%). During follow-up, eight patients (38%) suffered from recurrence or metastatic disease. All data are summarized in Table 1.

### 2.2. One Quarter of Patients Excluded after Blood Plasma Sample Quality Control

Blood samples were collected during the routine clinical workflow and then biobanked, and then quality assessment was performed before and after cfDNA isolation from plasma. Both samples from patient FR52 at baseline were hemolytic. For all other patients, neither duplicate was affected by hemolysis. The plasma samples from FR52 were not excluded because hemolysis does not affect bead-based cfDNA isolation methods [30]. Plasma samples were excluded if the sample volume was less than 2.5 ml (sum of volumes of duplicate blood samples), leading to the exclusion of seven patients. If, after cfDNA isolation, the plasma cfDNA amount was less than 10 ng, the sample was excluded unless the duplicate plasma sample yielded sufficient cfDNA. In the special case of follow-up plasma sample availability, all cfDNA samples were used regardless of cfDNA amounts. The sample exclusion results are summarized in Figure 1. For two of the twenty-one patients, we had follow-up cfDNA samples but no baseline cfDNA samples before neoadjuvant treatment, giving a total of nineteen patients with baseline cfDNA. All 21 patients’ tumor tissue samples were successfully sequenced.

### 2.3. Wide Genetic Diversity on the Level of Somatically Mutated Genes in EC Patients

Figure 2 and Appendix A summarize the somatic mutation spectrum and the mutation VAFs detected in the TT DNA for 20 of the 21 patients. For details on each patient see Appendix A. For the 21st patient’s TT specimen (FR04), no somatic mutation was detected by NGS. This was in agreement with the histological examination by the board-certified surgical pathologist, reporting complete tumor regression after neoadjuvant therapy. The most recurrently mutated genes were *TP53* (n = 16), *SMAD4* (n = 4), *TSHZ3* (n = 3), and *SETBP1* (n = 3). Most of the identified mutated genes (Figure 2) have been reported for EC before, but each study reported different sets of recurrently mutated genes, and no single EC study reported our list of recurrently mutated genes (Appendix A). The gene *SETBP1*, associated with histone modification [31], is mutated in three patients of our cohort and has not yet been reported in other EC studies. A relatively high TMB was found in two patients who received no neoadjuvant therapy (TMB_FR43_ = 15, TMB_FR52_ = 7.5) compared to the average TMB of 3.75 in our cohort. The most frequent protein-changing mutation type was a missense mutation (n = 49), followed by frameshift mutations (n = 7), and stop-gained mutations (n = 5). One loss of heterozygosity was detected in patient FR37 (Appendix A).

### 2.4. Lower Genetic Diversity on the Level of Signaling Pathways in EC Patients

A plot of cancer signaling pathways associated with the mutated genes in our patient cohort is depicted in Figure 3, demonstrating that several pathways were frequently affected in different patients. The most recurrently affected pathways were the p53 pathway (n = 16), followed by the transcription factor/regulator (n = 5), transforming growth factor beta (TGF-ß, n = 5), histone modifier (n = 5), and DNA repair (n = 5) pathways.

### 2.5. Highly Variable cfDNA Isolation Yields from Blood Plasma in EC Patients

The amount of cfDNA isolated from our patients’ blood samples was highly variable, ranging from 2 ng to 244 ng per blood tube. Figure 4 shows the variation for all baseline plasma cfDNA sample duplicates (averaged across sample duplicates). Details of the cfDNA concentrations in the elution buffer measured with Agilent Tapestation are displayed in Appendix A.

### 2.6. ctDNA Detected in 63% of Patients at Baseline with Median VAF of 0.4%

Figure 5 shows the VAFs of the most prominent somatic mutations detected in each patient at baseline. The sequencing depth of all samples is summarized in Appendix A, ranging from 1692 to 30,336× including the samples with too low cfDNA amounts. Before neoadjuvant therapy, ctDNA was detected in 12 out of 19 patients (63%), with a median VAF of 0.4% (Appendix A), considering the somatic mutation with the most prominent VAF in every patient.

The following patients presented unusual results worthy of special mention: For patient FR60 (UICC IV), all TT mutations were seen in FR60’s cfDNA (Appendix A), which was not generally the case for the other patients. For patient FR43 (UICC IV), we detected a VAF of 25.4% in *SF3B1* (Appendix A). However, in TT there was only a VAF of 2%. In contrast, in buffy coat DNA the VAF was 17%, suggesting a hematological origin such as clonal hematopoiesis of indeterminate potential (CHIP) and not a TT origin. For FR25 (Appendix A), no mutation was detected in baseline cfDNA, but the dominant TT mutation (at time of operation) in *TP53* was detected in cfDNA at 3.6% VAF at follow-up/progression. For FR31 (Appendix A), a *PIK3CA* mutation was detected in all seven cfDNA samples but not in TT or buffy coat. The origin of this *PIK3CA* mutation with a VAF of 4% is therefore unknown.

### 2.7. Longitudinal ctDNA Changes Are Consistent with Clinical Course

We sequenced longitudinal cfDNA samples from eight of the twenty-eight EC patients to explore its utility for monitoring the clinical course. The results are listed in Appendix A. Figure 6 shows the VAF changes in samples of three patients (FR04, FR13, FR52) taken before neoadjuvant treatment and at follow-up, in contrast to the progression of patient FR31 without neoadjuvant treatment. For patients FR04 and FR13, the cfDNA VAFs were reduced after neoadjuvant therapy (M4 in Figure 6). For patient FR52, who was in full remission after neoadjuvant therapy and operation, we sequenced the follow-up sample duplicate at three months after operation (M6 in Figure 6) and detected no cfDNA mutations. For patient FR31, who received no neoadjuvant therapy, the cfDNA VAFs increased and remained at a high level. The VAFs are consistent with the clinical courses of the patients:

Patient FR04 received four cycles of chemotherapy neoadjuvantly. Subsequently, the tumor was completely resected, and the resected specimen showed no involvement of the lymph nodes (R0 and N0). Follow-up by computed tomography (CT) 3 months postoperatively showed no signs of recurrence or distant metastases. However, a new CT check 7 months after surgery revealed a suspected peritoneal carcinomatosis. This was confirmed histologically and was derived from the EC.

Patient FR13 also underwent a complete resection without signs of lymph node involvement (R0 and N0). This was confirmed in the following follow-up examination. There were no signs of recurrence on CT 3, 7, and 10 months after surgery.

Patient FR31 underwent resection without prior neoadjuvant therapy due to the already stenosing EC. The surrounding diagnostics also revealed a suspicion of liver metastasis. The case was discussed in detail in the tumor conference and the indication for primary resection of the EC and subsequent resection of the liver metastasis in the course. After an initial complication-prone course with anastomosis insufficiency after esophageal resection, the liver metastasis resection was only performed 6 months after the initial esophageal surgery. However, follow-up CT at 14 months after esophagectomy showed radiological evidence for lung and renewed liver metastasis.

In patient FR52, after neoadjuvant chemotherapy and subsequent complete resection of the EC in accordance with the guidelines, the further postoperative course was unremarkable over the first few months [32]. Due to swallowing difficulties, a gastroscopy was performed in the ninth month postoperatively. This revealed only a stenosis at the anastomosis between the rest of the esophagus and the gastric tube. This was dilated endoscopically. However, the control endoscopy 12 months after the operation revealed a space suspected of malignancy, which was biopsied and revealed to be a local recurrence of the EC. In the course of further staging examinations, the CT also revealed suspected liver metastases, which were also confirmed histologically.

Further longitudinal results are shown for FR15, FR25, FR37, and FR54 in Appendix A, respectively. For patient FR25, a TT mutation was detected in follow-up plasma cfDNA samples but not in baseline plasma (Appendix A). TT mutations were also detected in the follow-up plasma samples from patient FR54, for whom there was no baseline sample. Here, the cfDNA VAF increased from 0.3% at follow-up 2 to 12% at follow-up 3 (Appendix A).

## 3. Discussion

To lower the high mortality rate in EC, therapeutic decision making, monitoring, and detection of therapy failure at the earliest time possible have to be improved. To date, no clinical liquid biopsy examination for monitoring the course of EC patients is established in routine practice. As previous studies provided conflicting evidence on detection rates of liquid biopsy and its utility in EC, our aim was to further investigate the utility of cfDNA for baseline examinations, dynamic monitoring of therapeutic responses, and detection of disease recurrence after R0-resection in patients suffering from EAC and ESCC. We therefore sequenced TT DNA and blood cfDNA samples from a cohort of northern German patients, using a Pan-Cancer panel that would fulfill the requirements for a routine clinical setting. This included using a commercial software for analysis. Our data from TT DNA and cfDNA analysis underlines the wide molecular diversity of EC cases that has previously been observed [18] and confirms the importance of liquid biopsy for improving clinical practice.

TT DNA sequencing from FFPE specimens is considered the gold standard for mutation detection. Examining the TT DNA, we found that TP53 was mutated in most of our patients (80%). This is in agreement with other sources and reports [18,24,33,34], detecting TP53 mutations in approximately 60% to 100% of patients. In addition to *TP53*, various other cancer genes were mutated among the patients (*SMAD4*, 20%; *TSHZ3*, 15%; *SETBP1*, 15%; and others with 10% or less). Strikingly, other studies only reported subsets of the genes that were mutated in our patients, with *SETBP1* not being found in our comprehensive literature research as an EC-related gene to date (Appendix A). This emphasizes that EC is characterized by a wide range of individual mutations depending on patient and cohort, necessitating a very large cohort to stratify patients by somatically mutated genes into potentially similar groups for potential treatment arms. We then speculated that potentially similar groups could be formed by considering cancer signaling pathways. Based on the cancer signaling pathways [35], we were able to narrow down the genetic diversity and identify more molecular recurrences between patients than on the gene level. The p53 pathway remained dominant (80% of patients), followed by the transcription factor/regulation (20%), TGF-ß (20%), histone modifier (20%), and DNA repair (20%) pathways. In 55% of our patients, the p53 pathway and at least one other recurrent pathway were affected by somatic mutations. Two patients had a high TMB, a biomarker for immunotherapy [36]. These two EAC patients received no neoadjuvant therapy, and it can only be speculated whether their TMB may have been lower if they had received neoadjuvant chemoradiation therapy (nCRT). Park and colleagues reported lower TMB in TT DNA after nCRT for EC patients [37]. Park’s observation may be specific for EC because genomic profiles in TT DNA were found to be unchanged in a study of rectal cancer patients under nCRT [38].

In this retrospective study design, samples were not preselected. All patients were included who consented to be included in the study. This allowed us to identify drop-out rates in real patient cohorts: amples with insufficient material for analysis were counted and reported. While our patient numbers were too small to perform a stratification, they were large enough to highlight the diversity of individual mutations that could be considered in a more individualized therapy. Since we wanted to dedicate ourselves in this work to the exploratory clarification of how many of the examined patients have detectable ctDNA and in which VAF range the mutations lie, we examined the two tumor histologies together. For a genomic stratification of EC, an initial stratification of EC patients into ESCC and EAC is usually performed and is sensible. Our work showed us that there is only a small overlap between different literature sources with respect to the most recurrently mutated genes, which in our view shows that EC is understudied and that further genomic studies with higher patient numbers from different populations and ethnicities are warranted.

The treatment protocols used in standard of care and in current clinical trials are consistent with the main cancer signaling pathways that we have identified in our cohort. Therefore, in future clinical trials or patient care it might be beneficial to match patient and protocol based on the affected pathways. The most widespread chemotherapy protocols in the treatment of EC are the FLOT and CROSS regimens [39]. In the FLOT regimen, chemotherapeutic treatment is given pre- and postoperatively, whereas in the CROSS regimen, treatment is exclusively neoadjuvant. With respect to pathways targeted by these regimens, it can be interpreted that they address a combination of the p53 pathway (exogenous stress leading to apoptosis) and proliferation pathways (taxanes act on dividing cells by blocking microtubule depolymerization during cell division [40]). With respect to the affected TGF-ß pathway, a new trial was recently launched: “Addition of TGF-β and PDL-1 Inhibition to Definitive Chemoradiation in Esophageal Squamous Cell Carcinoma (TAPESTRY)” (NCT04595149). With respect to the affected histone modifier pathway in our patients, a search of the clinicaltrials.gov website found the following trials: “Tucidinostat and PD-1 Inhibitor for Advanced Esophagus Cancer, AEG, Gastric Cancer” (NCT05163483), “A Phase I Study of LBH589 (Panobinostat) in Combination With External Beam Radiotherapy for the Treatment of Prostate Cancer, Esophageal Cancer and Head and Neck Cancer” (NCT00670553), and “Phase I Study of Gene Induction Mediated by Sequential Decitabine/Depsipeptide Infusion With or Without Concurrent Celecoxib in Subjects With Pulmonary and Pleural Malignancies” (NCT00037817) which include EC patients. This diversity of therapeutic options available or currently under investigation in clinical trials highlights the need for an easy and constantly accessible method to monitor changes in treatment response and then tailor and adjust the treatment of EC patients. Even though examining TT provides valuable insights into the tumor itself, it is not practical to take regular biopsies for monitoring treatment. Imaging methods are one option to monitor treatment response, and blood examinations are another option. Thus, blood sampling and sequencing of cfDNA can provide a scalable semi-automated platform for frequent monitoring of the clinical course.

In baseline cfDNA, we detected at least one mutation in twelve out of nineteen patients (63%). The median VAF at baseline was 0.4%. In all cfDNA samples (baseline or follow-up samples), we detected at least one mutation in fifteen out of twenty-one patients (71%). By comparison, previous EC studies detected mutations in cfDNA in 14% of patients with a small NGS panel that targeted regions from 12 genes [29], in 36% of patients with deep sequencing of the *TP53* gene [28], and in 60% of patients with a large NGS panel that targeted 802 regions from 607 ESCC/EAD genes [22]. Previous EC studies reported VAFs of 0.07% [22], 0.52% [25], 0.76% [24], and even up to 100% [26]. The high VAFs in the exome study [26] were only seen in library singletons and not replicated, and therefore they do not convince us given the much lower VAFs reported by studies [22], [24], [25], or found in our own study. Supporting our results are the experimental improvements that we made, e.g., duplicate blood draws, maximized cfDNA input amount, and a large NGS panel covering the genetic diversity of EC. In clinical practice, it will not be economic to run examinations that will only be successful in 14% or 36% of patients. In this respect, our study supports the studies with a higher detection rate, making clinical implementation more attractive. On the laboratory level, the different VAFs reported by these studies have practical implications. The current evidence shows very low cfDNA VAFs in EC patients, requiring very deep and expensive sequencing. As a second challenge, these VAFs are in the region of the signal noise levels of 0.1% to 0.5% [41] commonly seen using current sequencing methods. There are additional known confounders in cfDNA examinations which could account for differences between studies. Lampignano et al. benchmarked cfDNA extraction kits and showed that the VAF may depend on which kit is used [42]. Another study showed that the amount of cfDNA extracted can vary considerably in the same healthy person between different blood draws [30].

We performed peripheral venous blood sampling in Streck tubes to ensure the same pre-analytical conditions throughout the study because not all patients are equipped with a central venous port and because this port is usually removed after some time. Blood sampling via a central venous port system would have ensured that sufficient amounts of blood can be drawn and has been suggested to be beneficial for detecting ctDNA compared to peripheral venous blood vessels [43,44,45]. Peripheral access is often challenging due to patient compliance and vein conditions, and therefore some patients in the clinical routine setting will not be able to donate adequate blood samples for ctDNA monitoring. As a note, this was a prospective patient cohort study, and we are aware that new questions are raised that require further studies. For example, in our study, 25% of patients were excluded from the further workflow because there was less than 2.5 ml plasma in a 10 ml vacuum blood collection tube. It will also be worthy to investigate why some patient samples contained sufficient plasma volumes (3.5–5 ml plasma from a 10 ml vacuum tube) but yielded less than 10 ng of cfDNA, i.e., ca. 1500 genomes [46], while others yielded 10–20× as much cfDNA in the plasma sample.

As one of our most clinically relevant findings, it was possible to longitudinally monitor a decrease in ctDNA during therapy in the patients who received neoadjuvant therapy and an increase in the patient who did not (Figure 6 and Appendix A). This is an advance to the study by Ococks and colleagues, who were unable to demonstrate cfDNA-based monitoring with tumor regression [25].

As one of our key laboratory findings, it is noteworthy that the number of mutations that can be detected in the ctDNA of EC patients is usually lower than the number of mutations detected in corresponding TT. However, the high specificity of a tumor mutation and the rapid change of ctDNA levels in response to therapy make ctDNA an excellent biomarker [47]. To safeguard the specificity of a tumor mutation in cfDNA for disease monitoring, i.e., to rule out hematopoietic origins [48,49,50,51], we would recommend the inclusion of sequencing of the TT DNA and/or deep sequencing of leukocyte (buffy coat) DNA, as performed in our study and by others [52,53,54]. Therefore, future clinical practice should combine TT examinations with cfDNA examinations, possibly adding other well-studied blood markers such as *CK20* and *DEFA5* expression [55,56].

## 4. Materials and Methods

In total, the patient cohort consisted of 28 non-selected EC patients. All patients underwent surgical tumor resection in the Department of General Surgery, Visceral, Thoracic, Transplantation, and Pediatric Surgery, University Hospital Schleswig Holstein (UKSH), Campus Kiel, between November 2019 and March 2021. Patients with ESCC and EAC were included. This study was approved by the local ethics committee of the Medical Faculty, Kiel University (reference nos. A110/99, D615/21). All patients gave written informed consent before inclusion in the biobank for biomarker studies. Classification of the pathological tumor stage was conducted at the Department of Pathology, UKSH Campus Kiel, according to the TNM classification, 8th Edition [23]. Clinical data were obtained from the clinical research database of Translational Interdisciplinary Biobank Kiel (TRIBanK).

Details of the samples and laboratory and data analysis methods are provided in Appendix B. In brief, cancer tissue samples (n = 31) were obtained from resected primary tumor regions and metastases, and the unique pathology laboratory IDs were recorded in the laboratory information system (LIMS). Blood samples were collected during the routine clinical workflow and biobanked for this study. Samples were collected at presentation in the hospital (baseline) and at follow-ups. The blood samples were collected in Streck cfDNA BCT tubes (Streck, Inc., La Vista, NE, USA) in biological duplicates, 9 ml blood each. Duplicate blood samples were collected to identify and minimize artefacts in cfDNA analysis related to sample collection, processing, or NGS. Buffy coat and plasma were separated by centrifugation and stored frozen at −20 °C until further processing. The obtained plasma volume and hemolysis, if any, was recorded in the LIMS. DNA was isolated from the formalin-fixed paraffin-embedded (FFPE) tissue, blood plasma, and buffy coat using three specific kits. Correspondingly, three different NGS protocols were used. The IDT xGen Pan-Cancer panel v1.5 (Integrated DNA Technologies, Inc., Coralville, IA, USA) was used for enrichment of all samples. This is a capture-based panel that targets 127 cancer genes identified by The Cancer Genome Atlas (TCGA), totaling in a target region size of 800 kilobases. Sequencing was performed on NovaSeq 6000 (Illumina, Inc., San Diego, CA, USA). Scannable barcode stickers were used for all samples, tubes, and plates, and all steps were recorded in the LIMS.

Sequence data analysis was performed with GenSearchNGS (Phenosystems S.A., Braine le Chateau, Belgium) by comparing multiple samples (FFPE tissue samples in duplicates, buffy coat and plasma in duplicates, and commercial cfDNA reference samples, see Schema S27). We calculated the tumor mutational burden (TMB) based on TT DNA from the number of protein-changing somatic mutations per megabase. We considered only the variant calls with a population-based minor allele frequency of less than 1%. The recurrently mutated genes in the TT DNA were identified using a waterfall plot [57]. Relevant cancer pathways were assigned for each gene [35] and visualized in a pathway-based waterfall plot.

## 5. Conclusions

To address the continuously high mortality rate in EC, we suggest that molecular tumor classification be performed before therapy selection. Molecular classification should be performed by TT examinations because these provide a more reliable mutational profile than cfDNA. We resolved the conflicting data provided by different studies and therefore suggest that cfDNA should be used to monitor the clinical course. The ctDNA levels provide information on the tumor load and therefore could help refine the initial diagnosis and therapy decision. Longitudinal monitoring of cfDNA would allow for an individual adaptation of therapy over time. Future studies could investigate for which patients ctDNA examination could be beneficial for clinical decision making and disease monitoring. In conclusion, with the current methods, a combination of solid tissue and liquid biopsy samples should be used to help guide individualized EC therapy.

## Figures and Tables

**Figure 1 ijms-24-10673-f001:**
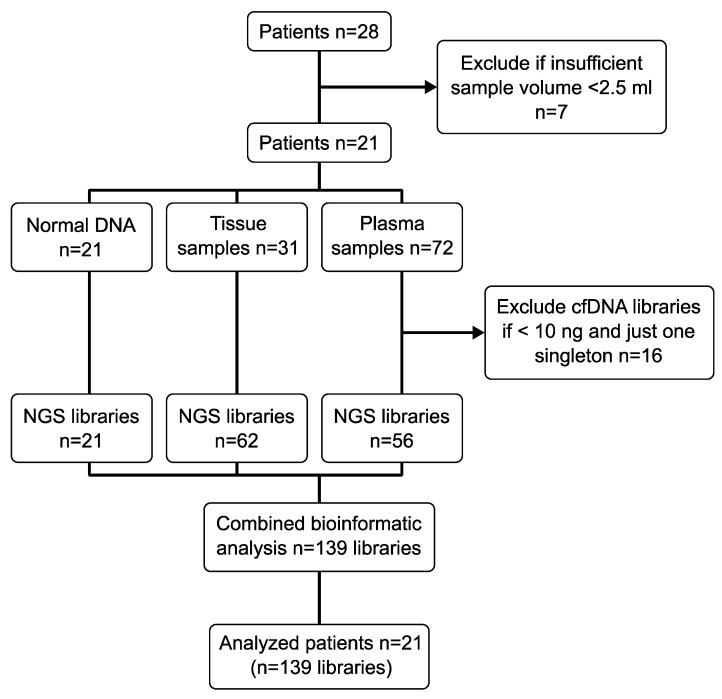
Consort diagram of EC patient samples and workflow.

**Figure 2 ijms-24-10673-f002:**
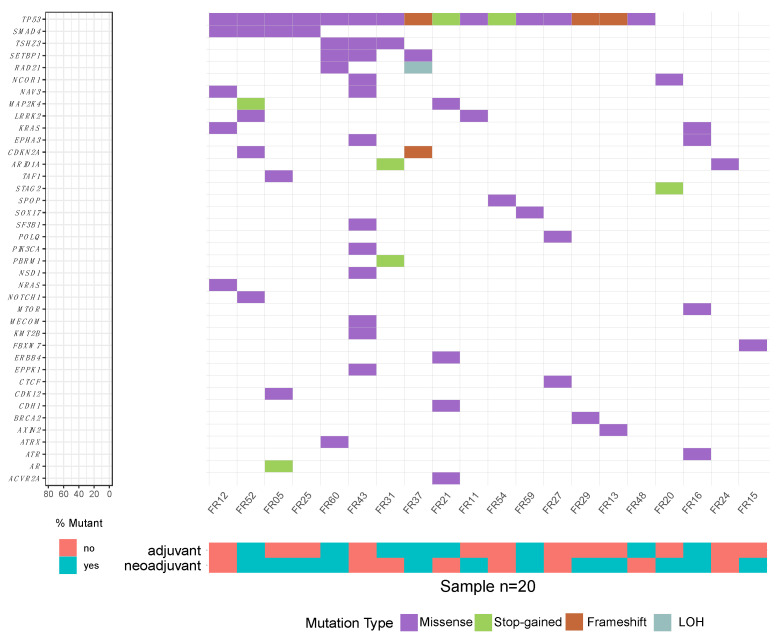
Oncoplot of detected somatic mutations and copy number changes in 20 EC tumors. The patients were sorted to group them by mutated genes.

**Figure 3 ijms-24-10673-f003:**
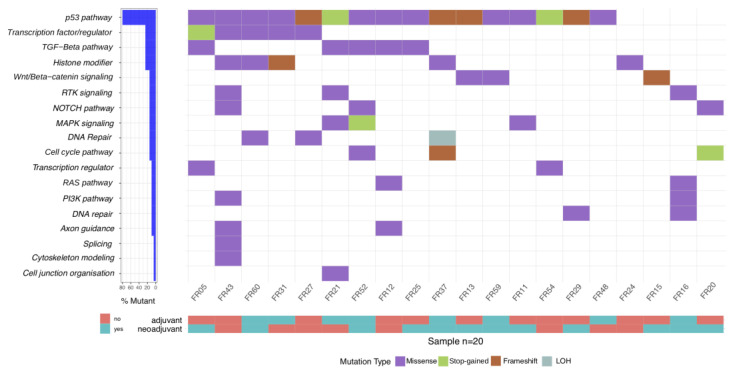
Oncoplot of affected pathways in 20 EC tumors, based on mutated genes. The patients were sorted to group them by affected cancer pathways.

**Figure 4 ijms-24-10673-f004:**
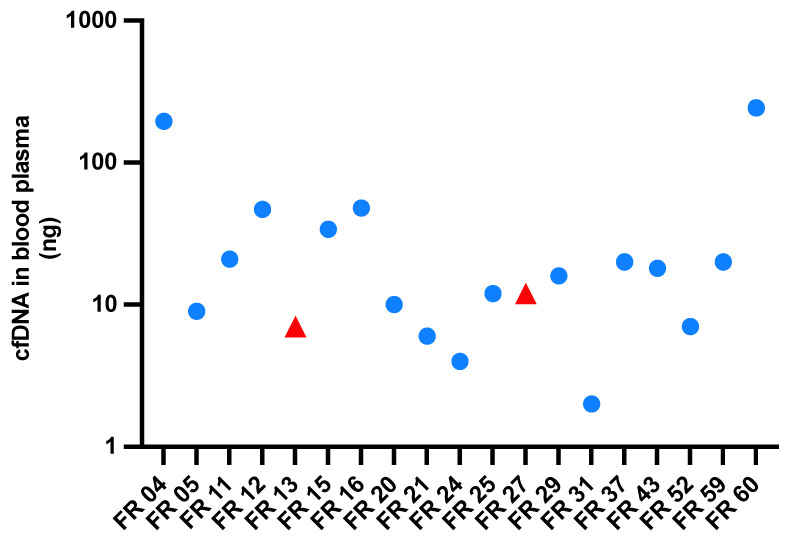
Amounts of cfDNA in baseline plasma (time of diagnosis, i.e., before neoadjuvant therapy or tumor resection) available for 19 of 21 EC patients. Patients with ESCC (FR13 and FR27) are marked with a red triangle. The patients with EAC are marked with a round blue dot.

**Figure 5 ijms-24-10673-f005:**
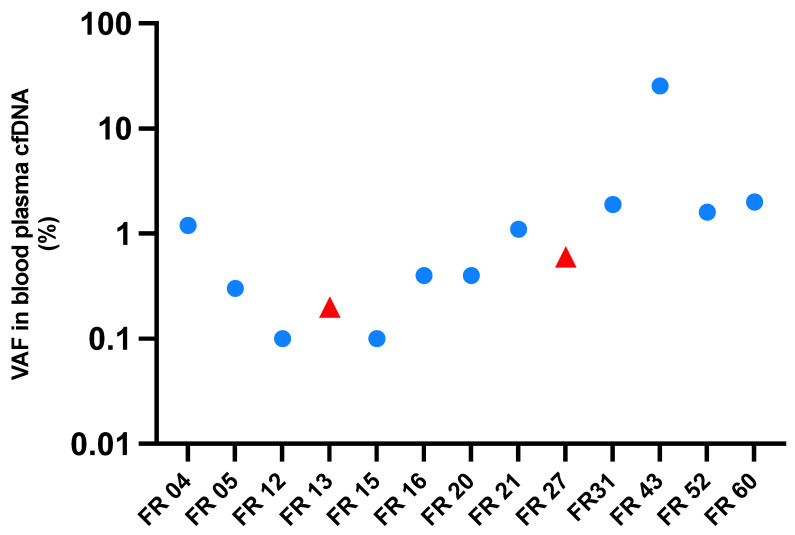
Mutant allele frequency in baseline plasma samples (time of diagnosis) detected in 13 of 19 EC patients. The figure shows the maximal variant allele frequency (VAF) of the most prominent somatic mutation detected in cfDNA. The maximal VAF was 25.4% (FR43). For FR11, FR24, FR29, FR37, and FR59 no confident somatic single nucleotide substitutions were detected in cfDNA, possibly reflecting low VAFs in the TT (7%, 2%, 1%, 7%, and 4%). Patients with ESCC (FR13 and FR27) are marked with a red triangle. The patients with EAC are marked with a round blue dot.

**Figure 6 ijms-24-10673-f006:**
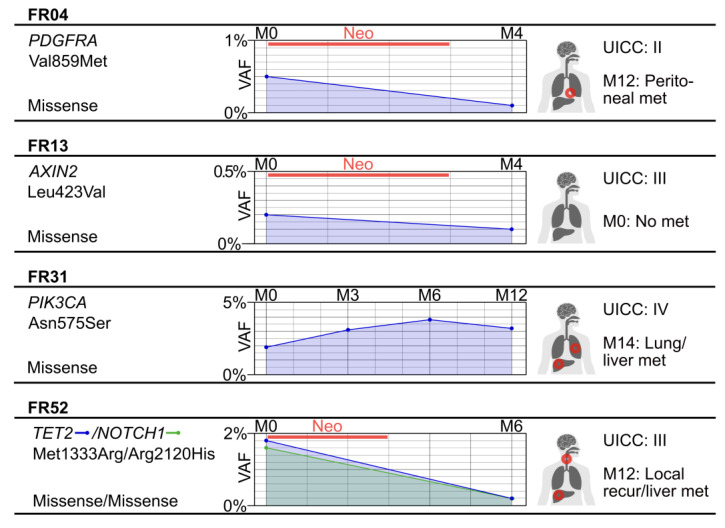
Longitudinal blood sampling of EC patients, their staging, and their clinical course. On the left side, the patient IDs, affected gene(s), and type(s) of displayed mutation are given. In the middle, the VAF of tumor mutations is given at the sampling times (M: month; M0, M4, etc.). The straight lines between sampling time points are drawn to facilitate comparison and do not represent the VAF between sampling time points. Only the most prominent mutation is depicted, except for patient FR52 with two similarly high VAFs in *TET2* and *NOTCH1*. The red bar (Neo) shows the length of neoadjuvant therapy. On the right side, the postoperative UICC classification and response and/or metastasis manifestation (met) are given.

**Table 1 ijms-24-10673-t001:** Patient demographics and clinical characteristics of the study population of EC patients.

		N (%)
**All**		**21 (100)**
Sex	Male	17 (81)
	Female	4 (19)
Age (years)	<70	10 (47.6)
	>70	11 (52.4)
Histological phenotype	EAC ^1^	19 (90.5)
	ESCC ^2^	2 (9.5)
pT category	pT1	2 (9.5)
	pT2	8 (38.1)
	pT3	11 (52.4)
pN category	pN0	7 (33.3)
	pN1	7 (33.3)
	pN2	3 (14.3)
	pN3	4 (19)
pM category	pM0	19 (90.5)
	pM1	2 (9.5)
UICC stage ^3^	I	1 (4.8)
	II	4 (19)
	III	10 (47.6)
	IV	6 (28.6)
Neoadjuvant treatment	Yes	13 (61.9)
	No	8 (38.1)
Adjuvant treatment	Yes	10 (47.6)
	No	11 (52.4)
Recurrence/Metastasis	Yes	8 (38.1)
	No	13 (61.9)

^1^ Esophageal adenocarcinoma. ^2^ Esophageal squamous cell carcinoma. ^3^ Union for International Cancer Control [23].

## Data Availability

The bam sequencing files were securely archived at the European Genome-Phenome Archive (EGA) under study accession ID EGAS00001006813.

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
