# Peer review of "Combining Solid and Liquid Biopsy for Therapy Monitoring in Esophageal Cancer"

_ijms, 2023, doi:10.3390/ijms241310673_

Round 1

Reviewer 1 Report

The "Combining solid and liquid biopsy for therapy monitoring in esophageal cancer" study is based on a fairly large and interesting NGS dataset, however, there is an inconsistent aspect in the description of the study that blurs the impression of this work, and I hope that the authors understand my remark. I want to say that the purpose of this study is stated as technical - to understand the discrepancy between the data of other works, however, the technical component of the study is discussed too little, instead, the mutations found (everything is as usual) and therapy (not investigated in this study) are discussed. It turned out that the data of some studies were confirmed, but why such strong discrepancies arose remained unclear, which means that they may arise again.

Now some minor notes:

Line 86: …analyses of EC TT…

The abbreviation TT should be defined.

Lines 119-125: This is a description of the results, which is not needed in the Introduction.

Figure 2 and 3: Why are 50 shades of pink used to represent different types of mutations? It is worth using different colors.

Line 182: Highly variable cfDNA isolation yields from blood plasma in EC patients

Here, it would be appropriate to present data on the control of hemolysis in these samples, especially since, as was shown in the FR43 sample, cfDNA contamination with blood cells DNA is quite possible.

Figure 6: I think it would be appropriate to enlarge this illustration to include results for patients FR15, FR25, FR37 and FR54 as well.

Lines 328-329: As one of our key clinical findings, it was possible to longitudinally monitor a decrease or increase in ctDNA, during therapy (Figure 6).

This is a rather controversial statement, since, as I understand it, it was possible to show this only on 3 patients. Perhaps this statement should be reformulated in such a way that its degree of proof is clear.

Appendix A - Method Details

1. For FFPE samples 2 x 100ng of DNA was used, for buffy-coat 100ng of DNA, and for cfDNA how much?

2. It should be given how hemolysis was assessed in a liquid biopsy study, because if there was hemolysis, then cfDNA is diluted with DNA from blood cells and this can affect the VAF. Perhaps this explains the different % in different studys, so this is an important point. If hemolysis assessment has not been done, then in my opinion this significantly reduces the value of this study.

Reviewer 2 Report

Dear authors,

Your manuscript, "Combining solid and liquid biopsy for therapy monitoring in esophageal cancer", shows results of evaluating somatic variations in circulating DNA (cfDNA and ctDNA) in esophageal cancer patients.  After you tested somatic mutations in cancer patients, you performed a liquid biopsy analysis looking forward to developing a monitoring technique to improve the follow-up of these patients.

Although the potential contribution of this study to its field, I would like to comment on some concerns:

Major comments

1. Please, add a statement regarding the statistical power of your sample. Would it be enough to reproduce your findings in other esophageal cancer cohorts?

2. Please, add detailed information about the quality of DNA obtained from all sources.

3. For Figures 2 and 3, I believe interesting to stratify patients who received neoadjuvant treatment. This pre-treatment could alter the number of somatic mutations. Also, these results (somatic mutations in tumors with or without neoadjuvant treatment) deserve further discussion.

4. For Figure 6. There is not clear if the monitored mutation in each case is always the same. If so, could you identify it? Add the proper variation next to the gene for each sample case. By the way, the FR13 patient decreased the number of copies mutated, which makes sense as this patient does not have metastases. But what about other cases with metastases? Could you discuss these results, please?

5. For this analysis, you used a systemic source of liquid biopsies. Could it be applicable to run this analysis on regional sources such as saliva? Are you planning to do it later? I think it is interesting to be discussed.

Minor comments

6. All classification guidelines must be cited. For example, UICC (line 136) or TNM system (line 360).

7. For all categorical asseverations, p-values could be included. For example, "The amount of cfDNA isolated from our patients' blood samples was highly variable (...) but comparable between sample duplicates"(lines 183-184). Have you compared cfDNA concentrations between paired blood tubes? If so, add the p-value.

8. For Figures 4 and 5. As y-axis ticks are also scaled, you must not need to include the (log10) term in the axis label.

Round 2

Reviewer 2 Report

Dear authors,

Your manuscript, "Combining solid and liquid biopsy for therapy monitoring in esophageal cancer", shows results of evaluating somatic variations in circulating DNA (cfDNA and ctDNA) in esophageal cancer patients.  After you tested somatic mutations in cancer patients, you performed a liquid biopsy analysis looking forward to developing a monitoring technique to improve the follow-up of these patients. Thank you for having answered my previous comments. Nevertheless, I still like to comment as follows.

Major comments

1. Please, add a statement regarding the statistical power of your sample. Would it be enough to reproduce your findings in other esophageal cancer cohorts? You mentioned that this study includes 21 patients: 19 with Esophageal adenocarcinoma (EAC) and 2 with Esophageal squamous cell carcinoma (ESCC). Given that these are cancers of different cell types, would it be correct to compare them combined? About tumor localization, do all patients belong to the same region in the esophagus? In esophagus-gastric tumors, there is known that esophageal tumors in the distal areas could be treated as gastric cancers. So, it could represent a difference in molecular terms.

2. Could you add the results of your samples after running the Agilent 2200 TapeStation with High Sensitivity D5000 Screen Tape (Agilent Technologies, Waldbronn, Germany), please?

Round 3

Reviewer 2 Report

Dear authors,

Your manuscript, "Combining solid and liquid biopsy for therapy monitoring in esophageal cancer", shows results of evaluating somatic variations in circulating DNA (cfDNA and ctDNA) in esophageal cancer patients.  After you tested somatic mutations in cancer patients, you performed a liquid biopsy analysis looking forward to developing a monitoring technique to improve the follow-up of these patients. Thank you for having answered my previous comments.